# Mind the GAP! The Challenges of Scale in Pixel-based Deep Reinforcement Learning

**Ghada Sokar**
Google DeepMind
gsokar@google.com

**Pablo Samuel Castro**
Google DeepMind
psc@google.com

## Abstract

Scaling deep reinforcement learning in pixel-based environments presents a significant challenge, often resulting in diminished performance. While recent works have proposed algorithmic and architectural approaches to address this, the underlying cause of the performance drop remains unclear. In this paper, we identify the connection between the output of the encoder (a stack of convolutional layers) and the ensuing dense layers as the main underlying factor limiting scaling capabilities; we denote this connection as the **bottleneck**, and we demonstrate that previous approaches implicitly target this bottleneck. As a result of our analyses, we present Global Average Pooling (GAP) as a simple yet effective way of targeting the bottleneck, thereby avoiding the complexity of earlier approaches.

## 1 Introduction

Reinforcement Learning (RL) is widely considered one of the most effective approaches for complex sequential decision-making problems [Mnih et al., 2015, Vinyals et al., 2019, Bellemare et al., 2020, Degrave et al., 2022, Wurman et al., 2022], in particular when combined with deep neural networks (typically referred to as deep RL). In contrast to the so-called "scaling laws" observed in supervised learning (where larger networks typically result in improved performance) [Kaplan et al., 2020], it is difficult to scale RL networks without sacrificing performance. There has been a recent line of work aimed at developing techniques for effectively scaling value-based networks, such as via the use of mixtures-of-experts [Obando Ceron* et al., 2024], network pruning [Obando Ceron et al., 2024], tokenization [Sokar et al., 2025], and regularization [Nauman et al., 2024]. Most of these techniques tend to focus on structural modifications to standard deep RL networks by leveraging sparse-network training techniques from the supervised learning literature.

Obando Ceron* et al. [2024] first demonstrated that naïvely scaling the penultimate (dense) layer in an RL network results in *decreased* performance, and proposed the use of soft mixtures-of-experts [SoftMoEs; Puigcerver et al., 2024] to enable improved performance from this form of scaling. Sokar et al. [2025] argued that the gains from SoftMoEs were mostly due to the use of tokenization. Relatedly, Obando Ceron et al. [2024] demonstrated that naïvely scaling the convolutional layers hurts performance, and showed that incremental parameter pruning yields gains that grow with the size of the original, unpruned, network. While effective, all these methods are non-trivial to implement and can result in increased computational costs.

One unifying aspect of the aforementioned works is that they tend to be most effective on networks that process pixel inputs, such as when training on the Arcade Learning Environment (ALE) [Bellemare et al., 2013]. These networks are typically divided into an *encoder* $\phi$ consisting of a series of convolutional layers, followed by a series of dense layers $\psi$; thus, for an input $x$, the network output is given by $\psi(\phi(x))$. Obando Ceron* et al. [2024] and Sokar et al. [2025] scaled the first layer of $\psi$ while Obando Ceron et al. [2024] scaled all layers in $\phi$. It is worth highlighting that $\psi \circ \phi$ is, in practice, a set of weights connecting the output of $\phi(x)$ with the input layer of $\psi$.

39th Conference on Neural Information Processing Systems (NeurIPS 2025).

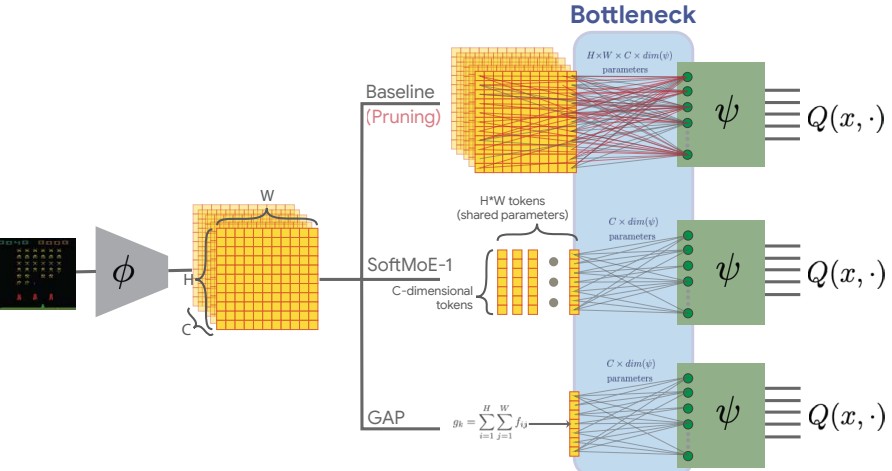

Figure 1: Illustration of the bottleneck in pixel-based networks. Standard dense networks (**Baseline**) connect all $\phi$ outputs with $\psi$, resulting in $H \times W \times C \times dim(\psi)$ parameters (scaled down when using **pruning**, shown in red). **SoftMoE-1** converts $\phi$'s outputs into $H \times W$ tokens of dimension $C$; the sharing of learned parameters across tokens results in a bottleneck with $C \times dim(\psi)$ parameters. **GAP** performs average pooling across $H \times W$ spatial dimensions, resulting in $C$ feature maps and $C \times dim(\psi)$ parameters in the bottleneck.

In this work, we argue that the underlying cause behind the effectiveness of the aforementioned methods is that they result in a **bottleneck** between $\phi(x)$ and $\psi$ (see Figure 1). As we will argue, performance gains are mostly due to a well-structured bottleneck, rather than the recently proposed architectural modifications to $\psi$. This insight suggests that simpler architectural interventions may be just as effective, which we demonstrate by using Global Average Pooling (GAP) as an effective mechanism for enabling improved performance from scaling.

Our contributions can be summarized as follows: (i) We investigate the challenges leading to the performance degradation of scaled RL networks; (ii) we study the underlying reasons behind the success of existing architectural approaches in scaling; and (iii) we present pooling as a faster, simpler method yielding superior performance. We begin in Section 2 by providing the necessary background on reinforcement learning, detailing the architecture employed in pixel-based deep RL, and outlining architectural modifications proposed by recent methods for addressing scaling challenges. Section 3.1 then describes our experimental setup. Our analyses investigating the difficulties of scaling RL networks and how existing methods attempt to address them are presented in Sections 3.2 and 3.3, respectively. Section 4 is dedicated to our results and analysis on RL networks with GAP. Finally, Section 5 covers the related work, followed by a conclusion and discussion in Section 6.

## 2 Preliminaries

### 2.1 Reinforcement learning

Reinforcement learning involves an agent moving through a series of states $x \in \mathcal{X}$ by selecting an action $a \in \mathcal{A}$ at discrete timesteps. After selecting action $a_t$ from state $x_t$, the agent will receive a reward $r_t(x_t, a_t)$, and its goal is to maximize the discounted sum of cumulative rewards $\sum_{t=0}^{\infty} \gamma^t r_t$, where $\gamma \in [0, 1)$ by finding an optimal *policy* $\pi : \mathcal{X} \rightarrow \Delta(\mathcal{A})$ which quantifies the agent's behavior at each state. Value-based methods [Sutton and Barto, 1998] maintain estimates of the value of selecting action $a$ from state $x$ and following $\pi$ afterwards: $Q(x, a) := \mathbb{E}\left[\sum_{t=0}^{\infty}\left[\gamma^t r_t(x_t, a_t)|x_0 = x, a_0 = a, a_t \sim \pi(x_t)\right]\right]$, where $\pi$ is induced from $Q$, for instance with the use of softmax: $\pi(x)(a) := \frac{e^{Q(x,a)}}{\sum_{a' \in \mathcal{A}} e^{Q(x,a')}}$.

Mnih et al. [2015] demonstrated deep neural networks can be very effective at approximating $Q$-values, even for complex domains such as Atari games [Bellemare et al., 2013]; their network has served as the backbone for most deep RL networks. Later, Espeholt et al. [2018] proposed

using ResNet based architecture which demonstrates significant performance improvements over the original convolutional neural network (CNN) architecture. For pixel-based environments, this family of networks consists of a set of convolutional layers [Fukushima, 1980], which we will collectively refer to as $\phi$ (and often referred to as the *encoder* or *representation*), followed by a set of dense layers, which we will collectively refer to as $\psi$. Thus given an input $x$, the network approximates the $Q$-values as $\tilde{Q}(x, \cdot) = \psi(\phi(x))$.

The output of $\phi$ is a 3-dimensional tensor $H \times W \times C$, where $H$ is height, $W$ is width, and $C$ is the number of feature maps (channels); this output is typically flattened before being fed to $\psi$. Thus, the number of parameters for the functional composition $\psi \circ \phi$ is equal to $H \times W \times C \times dim(\psi)$, where $dim(\psi)$ is the dimensionality of the first dense layer in $\psi$. We refer to connection between $\phi$ and $\psi$ as the **bottleneck**, and it will be the focus of most of our work. See Figure 1 for an illustration.

## 2.2 Network scaling

As we will demonstrate below, this bottleneck has a direct impact on learning efficiency. The standard approach is to flatten the output of $\phi$ into a single vector before feeding it to $\psi$, which we refer to as an 'unstructured' representation. This unstructured representation risks diluting the spatial structure captured by $\phi$, making learning more difficult. This is exacerbated when scaling the width of $\psi$, as demonstrated in prior works [Obando Ceron* et al., 2024, Sokar et al., 2025]. We hypothesize that the effectiveness of prior architectural modifications for scaling RL networks lies in their ability to induce some form of structure at this bottleneck. The rest of this section details these specific modifications.

### 2.2.1 SoftMoE

To address scaling limitations, Obando Ceron* et al. [2024] proposed a notable architectural change: replacing the standard dense layer in $\psi$ with a mixture-of-experts (MoEs). This required first restructuring the 3-dimensional output of the encoder ($\phi$) into a set of tokens to be fed into the mixture. Their investigation into tokenization strategies concluded that forming $H \times W$ tokens, each with $C$ dimensions (the channel depth), yielded the best performance. This restructuring effectively changes the connection, reducing the bottleneck's parameter count to $C \times dim(\psi)$ (See Figure 1). This architectural change has proven highly effective for various agents and at different scales.

Building on this, Sokar et al. [2025] isolated the source of these performance gains. They demonstrated that the tokenization step itself—not the MoE architecture—was the most critical component. Their evidence showed that a single expert model (SoftMoE-1), which retains the tokenization but omits the expert routing, achieves performance nearly identical to that of the full multi-expert model across different scales.

### 2.2.2 Sparse networks

Sparse methods offer an alternative approach to managing network complexity. Sokar et al. [2022] demonstrated that using sparse neural networks in place of dense ones can increase learning speed and lead to improved performance. Following this, Graesser et al. [2022] studied various ways to induce this sparsity, such as by pruning network weights during training or by using networks that are sparse from scratch. By imposing sparsity, these techniques directly structure the bottleneck, reducing the density of its connections. With a given sparsity level $s$, the effective number of parameters in the bottleneck is reduced to $s \times H \times W \times C \times dim(\psi)$. These sparse approaches have shown promise for the scaling of larger architectures; indeed, Obando Ceron et al. [2024] recently demonstrated that pruning is an effective approach that facilitates the scaling of networks.

**Gradual pruning**  This strategy begins with a standard, fully-dense network. As training progresses, connections (parameters) with low magnitudes—which are considered less salient to the network's function—are progressively removed. This pruning process often follows a polynomial schedule [Zhu and Gupta, 2017]. Once the pruning schedule concludes, the network achieves its target sparsity level, and this fixed sparse architecture is maintained for the remainder of training.

**Sparse from scratch**  In contrast to gradual pruning, this approach defines a sparse network at initialization, and this specific sparsity level is maintained throughout training. The sparse

topology can be *static*, meaning the set of active connections is fixed for the entire training duration. Alternatively, the topology can be *dynamic*. For example, the *RigL* method [Evci et al., 2020] dynamically optimizes the connections by periodically pruning a portion of the weights and growing new ones elsewhere, effectively "rewiring" the network while maintaining the overall sparsity.

## 3 Analyses

> **Main hypothesis**
>
> A low-density and well-structured **bottleneck** enables scaling deep RL networks.

We conduct a series of analyses, both quantitative and qualitative, to provide evidence for our main hypothesis. We investigate impacts on performance, plasticity, and properties of the learned features. Given the effectiveness of SoftMoEs [Obando Ceron* et al., 2024], Pruning [Obando Ceron et al., 2024], and Tokenization [Sokar et al., 2025] for scaling value-based networks, our analyses in this section will focus on these.

### 3.1 Experimental setup

**Architectures**  We employ the Impala architecture [Espeholt et al., 2018] for its superior performance, and while our analysis centers on it, we also confirm our findings' broad applicability using the standard CNN architecture [Mnih et al., 2015]. Our main experiments and analyses present results across various scaling factors for the width of $\psi$ (specifically, $\times 1$, $\times 2$, $\times 4$, and $\times 8$). When a scaling factor is not explicitly mentioned, our analysis defaults to the $\times 4$ scale. This is because, as shown in prior work [Obando Ceron* et al., 2024, Sokar et al., 2025, Obando Ceron et al., 2024], the $\times 4$ scale provides a substantial performance improvement, while further scaling tends to yield only marginal gains. Although layer width is our primary focus, we also include some preliminary investigations into scaling network depth in the appendix.

**Agents and environments**  Our primary experimental setup involves the Rainbow agent [Hessel et al., 2018] evaluated on the Arcade Learning Environment (ALE) suite [Bellemare et al., 2013]. For direct comparison with recent work, we use the same 20-game subset from Obando Ceron* et al. [2024] and Sokar et al. [2025]. However, we present our main results on the full ALE suite of 60 Atari games. We ran each experiment for a total of 200 million environment steps, with results averaged over $5$ independent seeds, except for the experiments with an increased replay ratio of 2, where we report the results at 50M steps. To further test the generality of our approach on discrete tasks, we also evaluate Rainbow on the Procgen benchmark [Cobbe et al., 2019]. Additionally, we assess performance on the data-efficient 100k benchmark [Łukasz Kaiser et al., 2020], for which we use DER, a version of Rainbow specifically tuned for that data-constrained setting. Finally, to demonstrate our findings extend beyond discrete action spaces, we also evaluate the Soft Actor-Critic (SAC) agent [Haarnoja et al., 2018] on continuous control tasks from the DeepMind Control (DMC) suite [Tassa et al., 2018].

**Code and compute resources**  For all our experiments, we use the Dopamine library[1] with Jax implementations [Castro et al., 2018]. For SoftMoE [Obando Ceron* et al., 2024], we use the official implementation integrated in Dopamine. For the sparse methods [Obando Ceron et al., 2024, Graesser et al., 2022], we use the same JaxPruner library[2] [Lee et al., 2024] used by [Obando Ceron et al., 2024]. All libraries have Apache-2.0 license. All experiments were run on NVIDIA Tesla P100 GPUs. The duration of each experiment ranged from 4 to 13 days, depending on the specific scale and algorithm. We present the exact run time for each case in Section 4.

**Implementation details**  For all algorithms, we use the default hyperparameters in the Dopamine library. For sparse-training algorithms, we follow [Graesser et al., 2022] and use 90% sparsity. Additional analysis covering other sparsity levels is included in the appendix. For gradual pruning, we start pruning at 8M environment steps and stop at 160M (80% into training), following the schedules

---

[1]Dopamine: https://github.com/google/dopamine
[2]JaxPruner: https://github.com/google-research/jaxpruner

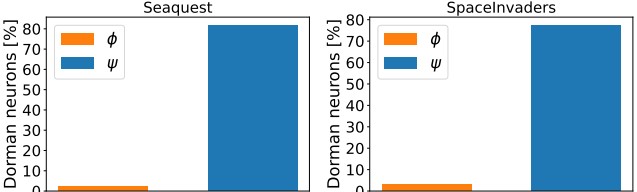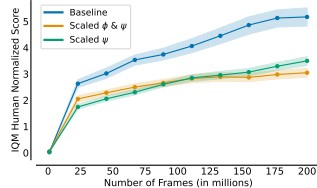

Figure 2: **(Left)** Distribution of dormant neurons across $\phi$ and $\psi$ in scaled baseline across different games at the end of training. The fully connected layer exhibits the highest percentage of dormancy. **(Right)** The performance degradation associated with scaling the entire network architecture is comparable to that observed when only the bottleneck is scaled. The performance is aggregated over 20 games.

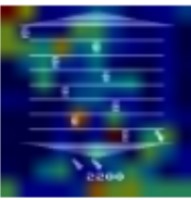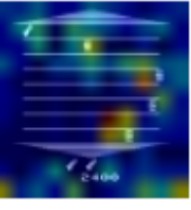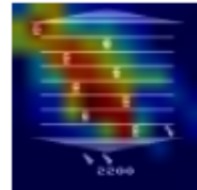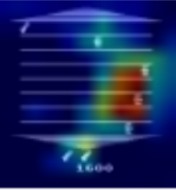

Figure 3: **GAP helps improve attention to relevant areas of input.** Visualizing influential regions for network decisions using Grad-CAM [Selvaraju et al., 2017]. **(Left)** The scaled baseline fails to attend to the important regions, focusing on irrelevant background details. **(Right)** GAP attends to the important regions in the input.

recommended by Graesser et al. [2022] and Obando Ceron et al. [2024]. For RigL [Evci et al., 2020], we use a drop fraction of 20% and a connection update interval of 5000. For SoftMoE, we use the single-expert variant (SoftMoE-1). This choice was twofold: first, it has been shown to yield performance comparable to the standard multi-expert SoftMoE [Sokar et al., 2025], and second, it ensures a more direct architectural comparison with the other methods considered in this study. We report interquartile mean (IQM) with 95% stratified bootstrap confidence intervals as recommended in [Agarwal et al., 2021]. Full experimental details are provided in the appendix.

### 3.2 Why scaling deep RL networks hurts performance

As has been previously demonstrated, naïvely scaling networks deteriorates performance [Obando Ceron* et al., 2024, Obando Ceron et al., 2024, Nauman et al., 2024]. In this section we conduct a series of experiments to diagnose the underlying causes for this difficulty.

We start by analyzing the training dynamics of a network where the width of all layers in both $\phi$ and $\psi$ are uniformly scaled by a factor of $4$. We examine neuron activity by measuring the fraction of dormant neurons, a metric that serves as a key indicator of network plasticity. Following Sokar et al. [2023], a neuron is considered "dormant" if its average activation falls below a certain threshold. As shown in the left plot of Figure 2, we find that $\psi$ exhibits a large fraction of dormant neurons, while $\phi$ has low dormancy rates. This suggests that **scaling mostly affects the plasticity of the bottleneck**.

In the right plot of Figure 2, we compare the performance of the baseline network against the same network with scaled $\phi$ and $\psi$. As consistently observed in previous studies, the scaled network exhibits a degradation in performance. To investigate the contribution of $\psi$ in this performance decrease, we only scale the bottleneck (via the first layer of $\psi$) to match the parameter count of the fully scaled network. As can be observed, the two scaled models has comparable performance, suggesting that **the bottleneck drives most of the performance degradation when scaling**.

To interpret the quality of the learned features of the scaled network, we generate saliency maps for the areas that have the greatest impact on the scaled network's output using Grad-CAM [Selvaraju et al., 2017]. Figure 3 (left) reveals that naïvely scaled networks fail to focus on important regions and focus on irrelevant background areas. This suggests that **scaling the bottleneck impairs a network's ability to process and learn effective combinations of the representation** $\phi$.

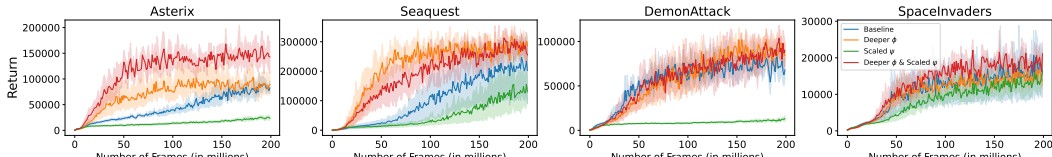

Figure 4: Scaling $\psi$ hinders learning effective combinations of encoder's features, leading to significant performance drop. However, performance dramatically improves when the scaled $\psi$ is fed with higher-level, more abstract features obtain by increasing the depth of $\phi$.

To further investigate potential challenges in feature learning within $\psi$, we study the network behavior when providing more abstract, higher-level features to $\psi$. Specifically, we increased the depth of $\phi$ by adding four additional ResNet blocks, while using the scaled bottleneck. This modification makes $\psi$ receive more structured, high-level features from the encoder. We present the performance throughout training in Figure 4. The fact that we observe a dramatic increase in performance when compared to the network that only scaled $\psi$ suggests that **structured representations helps feature learning in scaled networks.**

### 3.3 Existing techniques are mainly targetting the bottleneck

As previously mentioned, there have been a number of recent proposals to enable scaling deep RL networks by using sophisticated architectural modifications. We hypothesize that a core reason for their effectiveness is that they are implicitly targeting the bottleneck. The strong performance of SoftMoE-1 and tokenized baselines presented by Sokar et al. [2025] support this claim, as they are primarily re-structuring the encoder output.

To validate our hypothesis on sparse methods, we evaluated the impact of sparse training techniques when limited to the bottleneck, while keeping all other layers dense. We performed this analysis on gradual pruning [Obando Ceron et al., 2024], dynamic sparsity (RigL) and static sparsity [Graesser et al., 2022, Sokar et al., 2022]. As shown in the top plot of Figure 5, restricting sparsification to the scaled bottleneck results in improved performance across all sparse training techniques. This suggests that **applying sparsity only to the bottleneck is sufficient to enable scaling RL networks**.

In addition to structuring the encoder output, existing techniques effectively reduce the density of the bottleneck by either structuring the output as tokens (as in SoftMoE) which results in a lower dimensional input to $\psi$, or explicitly masking the majority of input weights (as in sparse methods). The bottom plot of Figure 5 confirms this reduction in parameters, and illustrates a positive correlation between scale and performance, which is notably absent in the baseline. This suggests that **low-density and structured bottlenecks facilitate scaling deep RL networks**.

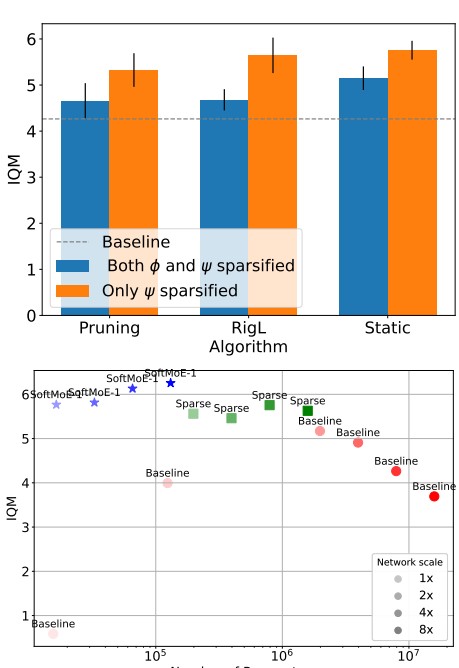

Figure 5: **(Top)** Across different sparse algorithms, sparsification of only $\psi$ yields better performance than sparsifying $\phi$ and $\psi$. **(Bottom)** The relation between performance and the effective number of parameters in $\psi$ for different approaches. Architectural methods have lower effective density than the baseline which correlates with the observed performance improvements.

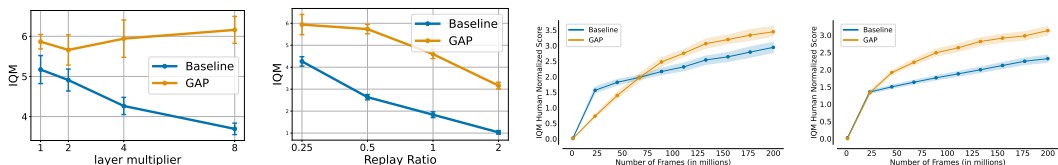

Figure 6: The impact of GAP across wide range of setting. From left to right: performance across different network scales, sample-efficient training with various high replay ratio values (default = 0.25), performance of the CNN architecture used by [Mnih et al., 2015], and performance on the full 60 games of Atari 2600. In all cases, GAP significantly improves performance.

## 4 Mind the GAP!

Having identified low-density and well-structured bottlenecks as the main factor enabling scaling networks, we demonstrate a *simple* alternative to the more sophisticated techniques recently explored: Global Average Pooling (GAP) [Lin et al., 2013]. We demonstrate its efficacy across wide range of settings and aspects, including: (1) *effectiveness* in network scaling under different scales, architectures, and in sample-efficient regime with high replay ratios (Figure 6); (2) *computational efficiency* (Figure 7); (3) improved training *stability* and feature learning (Figure 8); (4) unlocking *width and depth* scaling (Figure 9); and (5) *generalized gains* across various domains (Figure 11).

**Simple** The output feature maps of $\phi$, denoted as $F \in \mathbb{R}^{H \times W \times C}$, are processed by average pooling. For each feature map ($F^c$), GAP computes the average over its spatial dimensions, resulting in the output ($\mathbf{g} \in \mathbb{R}^C$), which is then fed to the fully connected layers $\psi$ (see Figure 1 for an illustration):

$$g^c = \frac{1}{H \times W} \sum_{i=1}^{H} \sum_{j=1}^{W} F_{ij}^c. \tag{1}$$

**Effective** We evaluate the impact of this architectural change on various settings. *Scale*: We assess the performance of GAP across different network scales. The left plot of Figure 6 demonstrates that this simple architectural change unlocks scaling in RL networks, leading to significant performance improvements across different scales. *Sample-efficient regime*: increasing the number of gradient updates per environment interaction (replay ratio) is favorable for sample-efficiency. Yet, higher replay ratios often hurt performance [Nikishin et al., 2022]. We study the performance of the scaled network across varying the replay ratio values (default = 0.25).

We find that even in this challenging setting, GAP has very strong performance compared to the baseline of the same network size, yielding more sample-efficient agents, as shown in the center left plot of Figure 6. *Varying architecture*: we evaluate the effect of GAP on a different architecture for $\phi$: the original CNN architecture used in [Mnih et al., 2015]. The right center plot in Figure 6 shows that GAP can also provide performance gain to this architecture. *Full suite:* we assess the generalization of our findings beyond the 20 games used for most results and evaluate on the full set of 60 games of Atari. The rightmost plot of Figure 6 confirms that GAP consistently improves performance on the full suite.

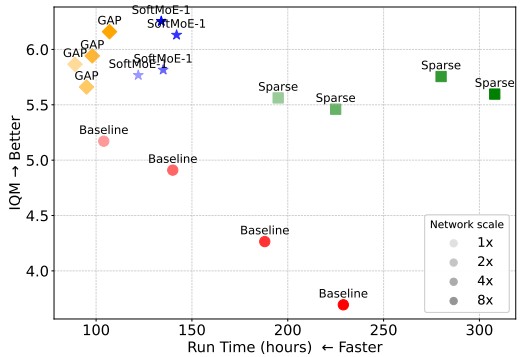

Figure 7: The computational cost versus performance across varying network scales for different algorithms. GAP offers the *highest* speed while obtaining substantial performance improvements.

**Efficient** In Figure 7 we report the total number of GPU hours per game for every scale, to compare the computational cost of the different methods. While scaling the baseline increases computational costs and degrades the performance, applying GAP to the encoder's output reduces

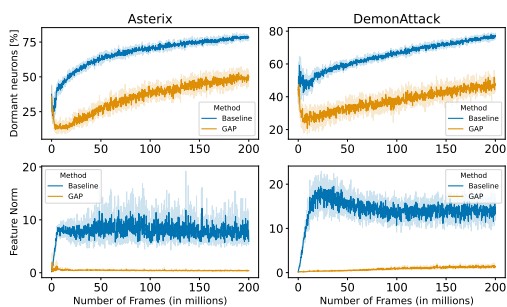

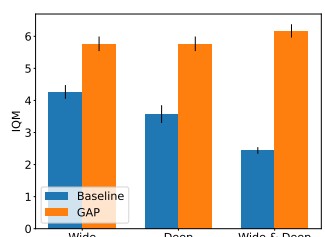

Figure 8: Scaled networks with GAP exhibit less dormant neurons than the baseline and have lower feature norm.

Figure 9: Impact of scaling network depth and width. Increasing depth hurts performance for the baseline networks, and scaling both width and depth makes it even worse. GAP, however, unlocks scaling even for increased network depth, leading to significant performance gains.

the effective density of $\psi$ (and hence computational cost), while yielding performance gains. Despite having a similar effective density to SoftMoE, GAP's inherent simplicity makes it more efficient by avoiding the extra computations associated with SoftMoE's token construction and post-MoE projection. The high runtime observed in sparse methods, despite their low effective density, stems from the fact that sparsity is only simulated with parameter masking [Hoefler et al., 2021, Mocanu et al., 2018, Evci et al., 2020].

**Stable training dynamics**   Figure 3 (right) displays the saliency maps when training with GAP, where the network seems to be attending to the most important and relevant areas of the input. We further examine the dormant neurons [Sokar et al., 2023] and the norm of the features in Figure 8. We observe that the network exhibits fewer dormant neurons than the baseline and lower feature norms, suggesting improved plasticity and training stability.

**Unlocking width-depth scaling**   We demonstrate the effectiveness of GAP in scaling network width which is the focus of previous works [Obando Ceron* et al., 2024, Obando Ceron et al., 2024, Sokar et al., 2025]. Beyond this, we also explore how increasing $\psi$'s depth, or both width and depth, impacts the performance of RL networks. Specifically, we add extra fully connected layer in $\psi$ and evaluate the performance with both unscaled and scaled width of $\psi$. More analysis with varying depth are included in the appendix. We present the results in Figure 9. We find that the baseline's performance drops with increased depth, worsening significantly when scaling both dimensions. In contrast, interestingly GAP maintains strong performance across all scaling dimensions, confirming the impact of the representation learned by the bottleneck in the overall performance of the network.

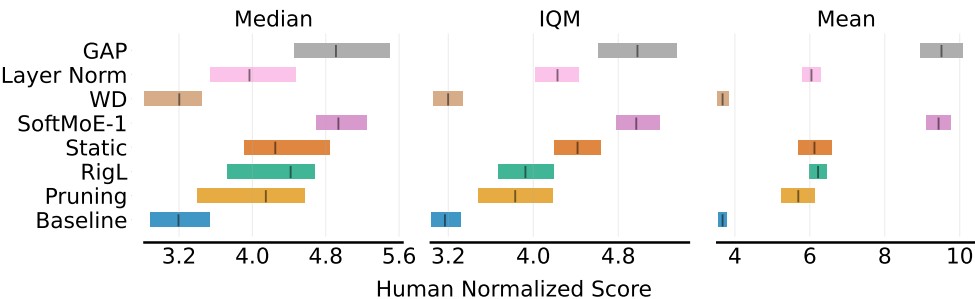

Figure 10: Comparison against architectural and algorithmic techniques for scaling RL networks. We report Median, IQM, and Mean scores [Agarwal et al., 2021] at 100M environment steps. GAP presents a notably simpler alternative to the baseline approaches, while achieving the best performance.

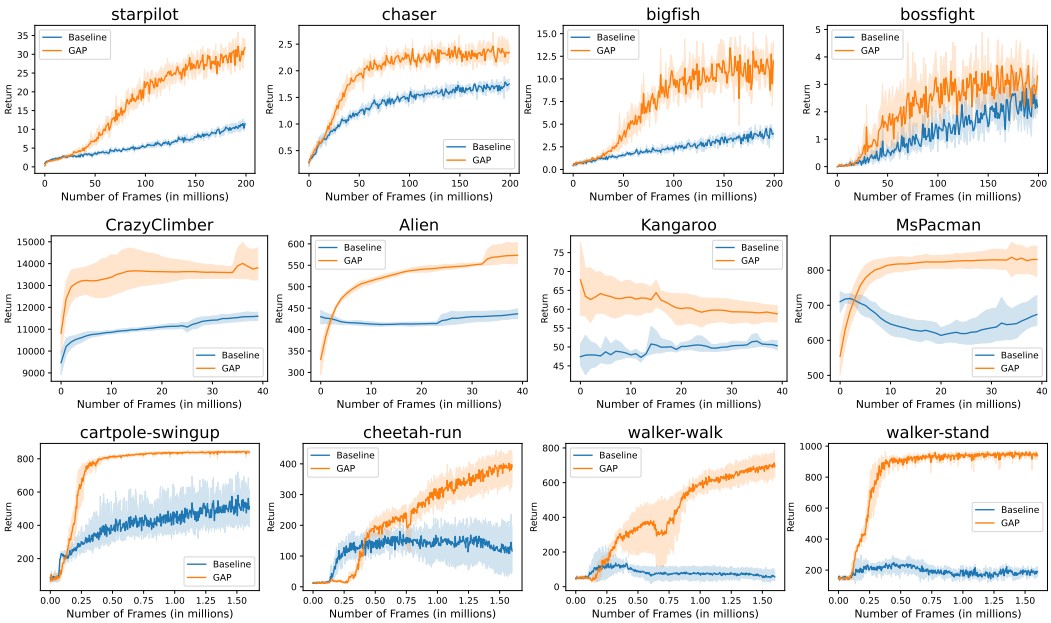

Figure 11: Performance for Rainbow on Procgen [Cobbe et al., 2019] **(top)**, DER on Atari100K [Łukasz Kaiser et al., 2020] **(middle)**, and SAC on DMC [Tassa et al., 2018] **(bottom)**. GAP leads to performance gains for scaled networks in diverse domains.

**Comparison against other methods**   We compare the performance of GAP against various architectural techniques including pruning [Obando Ceron et al., 2024], static and dynamic sparsity (RigL) [Graesser et al., 2022], and SoftMoE-1 [Sokar et al., 2025]. Moreover, we include a comparison against algorithmic methods including weight decay [Sokar et al., 2023, Obando Ceron et al., 2024], and layer normalization [Nauman et al., 2024]. Figure 10 shows that GAP outperforms all baseline methods and achieves performance comparable to SoftMoE, while being more efficient and simpler. Although GAP is a more aggressive compression technique than granular approaches like per-conv tokens in SoftMoEs [Obando Ceron* et al., 2024], which could lead to a loss of spatial structure, it still preserves spatial information within each feature map more effectively than a flattening operation.

**Generalization to other domains**   To further assess the broad applicability of our findings, we extended our evaluation to a diverse set of domains and agents. We assessed the Rainbow agent on Procgen [Cobbe et al., 2019], the DER agent [Van Hasselt et al., 2019] in the data-efficient Atari100K setting [Łukasz Kaiser et al., 2020], and the SAC agent [Haarnoja et al., 2018] on the continuous DeepMind Control (DMC) suite [Tassa et al., 2018]. For SAC, we follow the CNN architecture presented in [Yarats et al., 2021b] and scale the embedding layer of the actor and critic by 8. Figure 11 illustrates that these experiments align with our main results, confirming that GAP provides a consistent and significant performance benefit for scaled networks across these distinct benchmarks.

## 5   Related Work

Several works have shown that scaling RL network causes substantial performance degradation due to training instabilities exhibited by the network [Hessel et al., 2018, Bjorck et al., 2021, Obando Ceron* et al., 2024]. Although the precise causes of these issues remain unclear, several approaches aim to mitigate them. We categorize these methods to architectural methods that alter the standard network architecture and algorithmic methods.

**Scaling through architectural changes**   Recent works have investigated architectural modifications to improve the scaling of RL networks. Obando Ceron* et al. [2024] integrated a Mixture-of-Experts (MoE) after the encoder in single-task RL networks. Their results across multiple domains

highlight the effectiveness of this approach for scaling RL, with SoftMoE [Puigcerver et al., 2024] outperforming traditional MoE Shazeer et al. [2017]. This MoE approach was later extended by Willi* et al. [2024], who demonstrated its applicability in the multi-task setting. Relatedly, Sokar et al. [2025] provided insights into why such methods might succeed, showing that replacing the standard flattened representation from the encoder with a tokenized representation that preserves spatial structure significantly improves the performance of scaled networks. Another line of research studies replacing dense parameters by sparse ones in both online [Graesser et al., 2022, Tan et al., 2023, Sokar et al., 2022] and offline RL [Arnob et al., 2021]. This approach has demonstrated its effectiveness in increasing the learning speed and performance of RL agents. Network sparsity is achieved either by starting with a dense network and progressively pruning weights, or by initializing with a sparse network and maintaining a consistent sparsity level throughout training. In the latter case, the sparse structure can be kept static or optimized dynamically during training using methods like SET [Mocanu et al., 2018] and RigL [Evci et al., 2020]. Recently, Obando Ceron et al. [2024] showed that gradual magnitude pruning helps in scaling RL networks, leading to improved performance. Concurrent works have independently proposed similar approaches to GAP [Trumpp et al., 2025, Kooi et al., 2025], which provides extra evidence for the efficacy of this method.

**Scaling through algorithmic changes**   A primary goal in this line of research is maintaining training stability as networks grow. Bjorck et al. [2021] show that using spectral normalization [Miyato et al., 2018] helps to improve training stability and enable using large neural networks for actor-critic methods. Farebrother et al. [2023a] explore the usage of auxiliary tasks to learn scaled representations. Farebrother et al. [2024] demonstrated that training value networks using classification with categorical cross-entropy, as opposed to regression, leads to better performance in scaled networks. Other regularization techniques have been shown to be critical. Schwarzer* et al. [2023] propose several tricks to enable scaling including weight decay, network reset [Nikishin et al., 2022], increased discount factor, among others. Similarily, Nauman et al. [2024] employ a combination of layer normalization [Ba et al., 2016], weight decay, and weight reset.

## 6   Conclusion

Recent proposals for enabling scaling in deep reinforcement learning have relied on sophisticated architectural interventions, such as the use of mixtures-of-experts and sparse training techniques. We demonstrated that these methods are indirectly targeting the **bottleneck** connecting the encoder $\phi$ and dense layers $\psi$, in a standard deep RL network. A consequence of our analyses is that directly targeting this bottleneck, for instance with global average pooling, can achieve the same (or higher) performance gains.

Our work highlights the importance of better understanding the training dynamics of neural networks in the context of reinforcement learning. The fact that a simple technique like global average pooling can outperform existing literature suggests that architecture design is ripe for exploration. There is also a likely connection of our findings with works exploring representation learning for RL [Castro et al., 2021, Kemertas and Aumentado-Armstrong, 2021, Zhang et al., 2021, Farebrother et al., 2023b], given that these generally target the output of $\phi$; indeed, most of these methods aim to *structure* the outputs of $\phi$ so as to improve the generalizability and efficiency of the networks. It would be valuable to investigate whether approaches like GAP are complementary with (and ideally help enhance) more sophisticated representation learning approaches.

**Limitations**   Although our broad set of results suggest our findings are quite general, our investigation has focused on pixel-based environments where there is a clear bottleneck, or separation between $\phi$ (the convolutional layers) and $\psi$ (the fully connected layers). It is not clear whether our findings extend to non-pixel based environments or architectures where there isn't a clear bottleneck, but would be an interesting line of future work.

## Acknowledgements

The authors would like to thank Gheorghe Comanici, Joao Madeira Araujo, Karolina Dziugaite, Doina Precup, and the rest of the Google DeepMind team, as well as Roger Creus Castanyer and Johan Obando-Ceron, for valuable feedback on this work.

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

# A    Broader impacts

This paper studies the challenges in scaling networks in pixel-based deep reinforcement learning. We present a simple, effective alternative to current sophisticated approaches by identifying the effective network representation. The approach has a positive impact by requiring no hyperparameters, simplifying implementation and showing robust performance across various scenarios. While this research aims to advance RL agent capabilities without direct negative impact, we urge careful consideration of potential implications when building upon this work.

# B    Experimental Details

Table 1: Hyper-parameters for Rainbow and DER agents.

|  | | Atari | |
|  | Hyper-parameter | Rainbow | DER |
| --- | --- | --- | --- |
| Training | Adam's ($\epsilon$) | 1.5e-4 | 0.00015 |
|  | Adam's learning rate | 6.25e-5 | 0.0001 |
|  | Batch Size | 32 | 32 |
|  | Weight Decay | 0 | 0 |
| Architecture | Activation Function | ReLU | ReLU |
|  | Fully connected layer Width | 512 | 512 |
| Algorithm | Replay Capacity | 1000000 | 1000000 |
|  | Minimum Replay History | 20000 | 1600 |
|  | Number of Atoms | 51 | 51 |
|  | Reward Clipping | True | True |
|  | Update Horizon | 3 | 10 |
|  | Update Period | 4 | 1 |
|  | Discount Factor | 0.99 | 0.99 |
|  | Exploration $\epsilon$ | 0.01 | 0.01 |
|  | Sticky Actions | True | False |

**Hyperparameter details**   We use the default hyperparameters for all the studied algorithms. We present the values of these parameters in Table 1. For the dormant neuron analysis, we use a dormancy threshold of 0.001. For the feature learning analysis (Figure 4), we increase the depth of the encoder by adding two ResNet blocks.

**Atari Games [Bellemare et al., 2013]**   We use the set of 20 games used in Sokar et al. [2025], Obando Ceron* et al. [2024] for direct comparison. The set has the following games: Asterix, SpaceInvaders, Breakout, Pong, Qbert, DemonAttack, Seaquest, WizardOfWor, RoadRunner, Beam-Rider, Frostbite, CrazyClimber, Assault, Krull, Boxing, Jamesbond, Kangaroo, UpNDown, Gopher, and Hero. This set is used in most of our analysis, nevertheless we provide our main results on the full suite of 60 games.

**Atari100K Games [Łukasz Kaiser et al., 2020]**   We test on the 26 games of this benchmark. It includes the following games: Alien, Amidar, Assault, Asterix, BankHeist, BattleZone, Boxing, Breakout, ChopperCommand, CrazyClimber, DemonAttack, Freeway, Frostbite, Gopher, Hero, Jamesbond, Kangaroo, Krull, KungFuMaster, MsPacman, Pong, PrivateEye, Qbert, RoadRunner, Seaquest, UpNDown.

# C    Extra Experiments

**Sparse methods address the bottleneck**   Extending our analysis in section 3, we validate our hypothesis on various sparsity levels. Consistent with our main results, sparsification of only $\psi$ yields better performance than sparsifying $\phi$ and $\psi$ across all sparsity levels as shown in Figure 12.

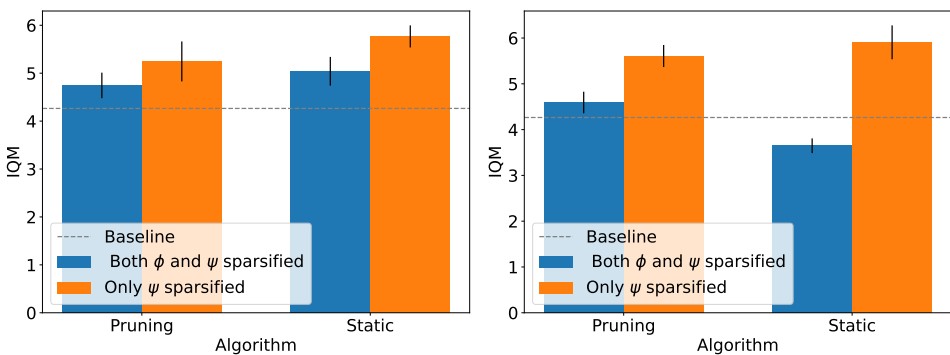

Figure 12: Sparsification of $\psi$ yields better performance than sparsifying $\phi$ and $\psi$ for 80% sparsity **(left)** and 95% sparsity **(right)**.

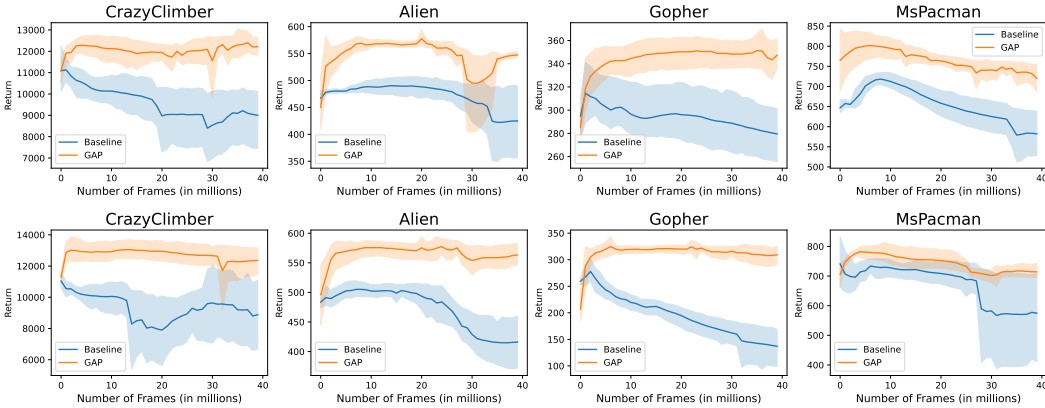

Figure 13: Performance for DrQ **(top)** and DrQ($\epsilon$) **(bottom)** [Yarats et al., 2021a, Agarwal et al., 2021] on Atari100K.

**More agents**  We evaluate two more algorithms in the sample-efficient regime: DrQ and DrQ($\epsilon$) [Yarats et al., 2021a, Agarwal et al., 2021] on Atari100K. Consistent with our main results, GAP improves performance of scaled networks as shown in Figure 13.

**Comparison against max pooling**  We compare the performance of global average pooling with global max pooling for Rainbow on the set of 20 games used in our main results. We find the GAP outperforms max pooling as demonstrated in Figure 14. We hypothesize this is due to GAP's ability to retain more comprehensive information through its averaging operation.

**Encoder width scaling**  We perform additional experiment analyzing the effect of scaling the width of layers in $\phi$ by a factor of 4, while maintaining $\psi$ unscaled. Figure 15 shows that GAP significantly improves performance.

**Deeper networks**  While our main focus in this work is scaling the width of the network, we also performed some preliminary analysis on scaling the depth of the fully connected layers ($\psi$), exploring architectures with 1, 2, and 3 additional layers. As shown in Figure 16, increasing the depth degrades the performance of the agent. Increasingly, GAP improves performance over the corresponding baseline network of the same size across varying depth.

**Performance throughout training**  Figure 17 and Figure 18 present the performance per game throughout training for Atari and Atari100K benchmarks, respectively.

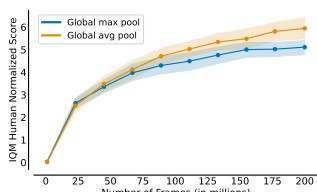

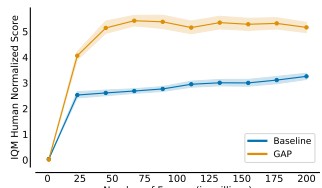

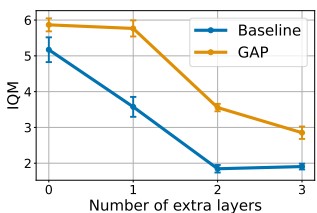

Figure 14: Comparison between GAP and global max pooling for Rainbow.

Figure 15: Effect of increasing the width of $\phi$. GAP improves performance over the scaled baseline.

Figure 16: Effect of increasing the depth of $\psi$. GAP achieves better performance than the corresponding baseline with the same network size.

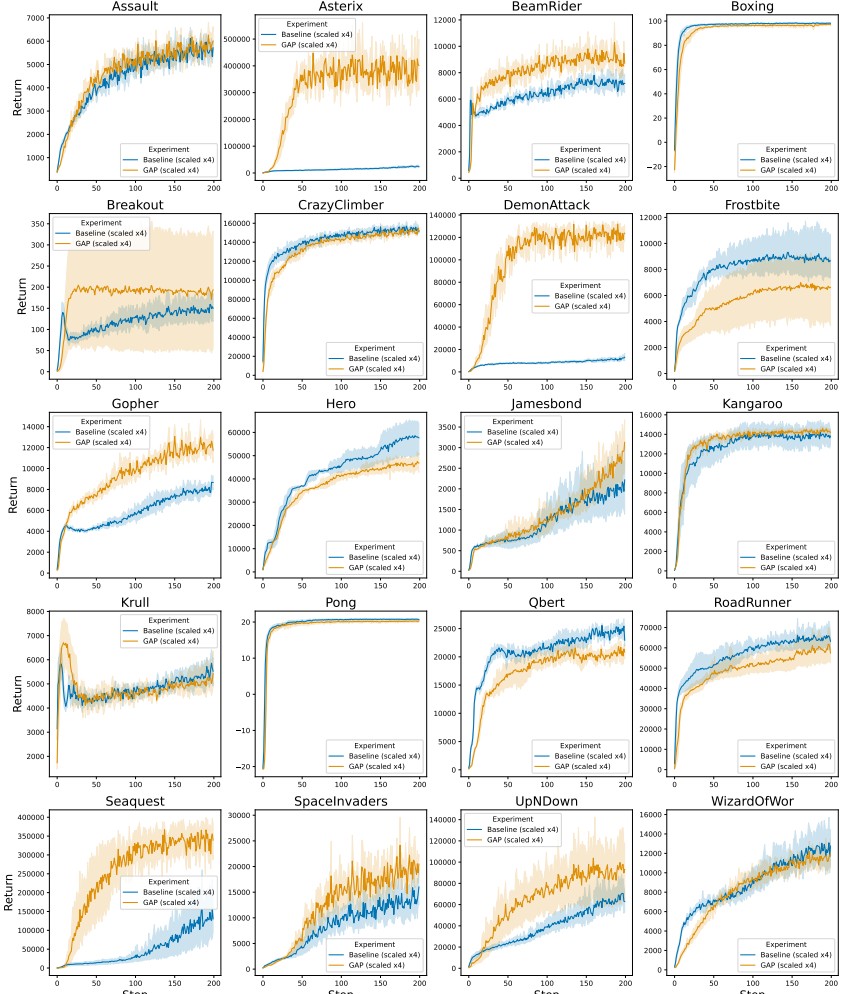

Figure 17: Performance during training on the 20 games of the Atrai benchmark.

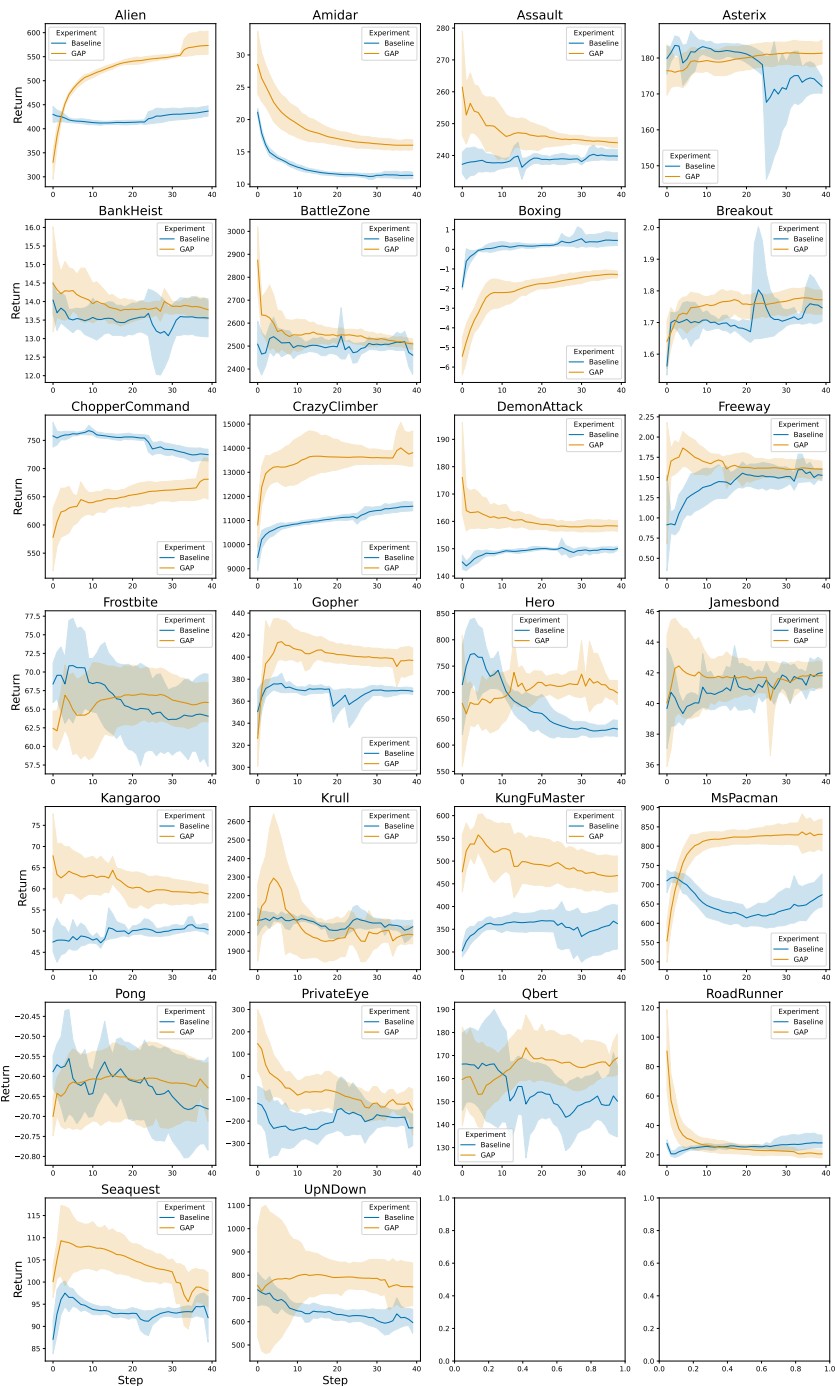

Figure 18: Performance during training on the 26 games of the Atrai100k benchmark.

