# OpenReview forum: "Mind the GAP! The Challenges of Scale in Pixel-based Deep Reinforcement Learning"
_NeurIPS.cc/2025/Conference — NeurIPS 2025 poster_

### Official Review · Reviewer_iWUt · 2025-06-28

**Clarity:** 3
**Significance:** 4
**Originality:** 4
**Rating:** 5
**Confidence:** 4

**Summary:**

This paper investigates the reasons behind the performance drop observed when scaling deep reinforcement learning (RL) in pixel-based environments. The authors pinpoint the connection between the convolutional encoder's output and the subsequent dense layers, which they term the "bottleneck", as the primary limiting factor. They argue that previous successful approaches implicitly address this bottleneck. Based on their analysis, the paper proposes Global Average Pooling (GAP) as a straightforward and effective method to target this bottleneck, offering a simpler alternative to more complex solutions found in the literature.

**Questions:**

It is currently unclear how 'dormant' neurons are determined. There are numerous possible approaches (e.g., activity thresholds, gradient magnitudes, L0 norm of weights). Please explicitly define the methodology used to identify 'dormant' neurons. If multiple approaches were considered, briefly comment why the chosen method was selected over others.

You mention following the work of others and aiming for 90% sparsity. The effect of increased sparsity on model changes and performance is a crucial aspect that needs further elaboration. Can you provide more details on how sparsity was implemented (e.g., pruning method, frequency). Elaborate on the observed effects of varying sparsity levels (even if only 90% was the target) on both the model's architecture (e.g., changes in connectivity, effective parameters) and its performance metrics.

The scaling by 4 factor seems arbitrary without further context, though there is likely a good reason for it. Is it related to a specific architectural choice, computational constraint, or a baseline from previous work?

Figure 10 & Performance Claims. You conclude that "the performance of the 2 scaled models is nearly identical." However, observation of the IQRs (presumed to be the shaded regions) shows non-overlapping distributions across many frame numbers. If the scaling of this figure were different, these deviations might appear more significant. While the overall conclusion regarding the bottleneck might be correct, it's important to acknowledge and comment on these visible deviations in the IQRs. If the differences are not considered significant, please explain why.

Overlapping distributions for GAP and SoftMoE are shown [boxplots]. It's questionable whether a notched boxplot or a Wilcoxon rank-sum test would show statistically significant differences in their medians. Given this visual overlap, can you provide a more robust argument for why GAP is considered "higher performing than all other methods." If statistical tests were performed, please give the results. If not, consider performing them or refining the claim.

Minor Edits
P3 Deep RL network > networks: Change "network" to "networks" for plural consistency.
P3 tokenzing > tokenizing: Correct the spelling from "tokenzing" to "tokenizing."
Fig 11 ProcGen [spelling]: Correct the spelling of "ProcGen" in the figure caption

**Ethical Concerns:**

["NO or VERY MINOR ethics concerns only"]

**Final Justification:**

the author responses to the questions raised were thorough - I am keeping my score of 5 'accept'

**Limitations:**

yes

**Quality:**

3

**Strengths And Weaknesses:**

Strengths
The paper offers a simple, effective, and practical solution to a common bottleneck in supervised learning, particularly relevant to RL pixel literature. The authors have conducted a thorough empirical investigation with commendable breadth and presentation of experiments. A key strength is the demonstration that "bigger doesn't mean better" in many RL problems, providing an easily implementable fix. The inclusion of wall-clock times for training versus performance is a valuable addition, highlighting the practical benefits of their approach. The application of sparsity to the bottleneck is a compelling finding, with the results presented being impressive and supporting the method's benefits. The core idea of investigating 'dormant' neurons and their impact on model performance is also a useful insight.

Weaknesses
Despite its strengths, the paper has several areas that need improvement.
Figure 9 is unclear and needs enhancement to better convey its message.
While empirical evidence is strong, the paper would benefit from theoretical backing for its claims.
Training curves for all baseline methods, not just the full (non-GAP) network, are needed for a comprehensive comparison.
The methodology for determining 'dormant' neurons is unclear and needs explicit definition, along with a justification for the chosen approach.
More details are required on how sparsity was implemented (e.g., pruning method, frequency) and the observed effects of varying sparsity levels on model architecture and performance.
The claim that "the performance of the 2 scaled models is nearly identical" is questionable given the non-overlapping IQRs in Figure 10. A more nuanced discussion or explanation for why these deviations are not significant is needed.
The claim that GAP is "higher performing than all other methods" lacks robust statistical justification, especially given the visual overlap in distributions. Statistical test results (e.g., Wilcoxon rank-sum test) are needed to support this claim, or it should be refined.

---

> ### Author Rebuttal · Authors · 2025-07-31
>
> We thank the reviewer for their feedback! We are glad that the reviewer found the method simple and effective and that the empirical investigation is thorough and strong.
>
> > **Q1:** It is currently unclear how 'dormant' neurons are determined.
>
> **A:** We adopt the definition from the original paper that identifies dormant neurons in deep reinforcement learning [1]. The dormancy is measured based on a neuron's activation threshold. We will add the precise formulation to the revised manuscript.
>
> > **Q2:** You mention following the work of others and aiming for 90% sparsity. The effect of increased sparsity on model changes and performance is a crucial aspect that needs further elaboration. Can you provide more details on how sparsity was implemented (e.g., pruning method, frequency). Elaborate on the observed effects of varying sparsity levels (even if only 90% was the target) on both the model's architecture (e.g., changes in connectivity, effective parameters) and its performance metrics.
>
> **A:** In the case of static sparsity, a random network topology with the specified sparsity level is initialized at the beginning of training and kept fixed throughout. In the case of pruning, we followed the same strategy of prior work that studied pruning in RL [2,3]. The network starts with a dense network and low magnitude parameters are gradually pruned throughout training according to a polynomial schedule. We will provide more experimental details of these analyses in the revised version.
>
> Following the reviewer’s suggestion, we have conducted extra experiments varying the sparsity levels. We report the IQM with 95% confidence interval at 40M steps aggregated over the 20 studied games. Consistent with our previous findings, sparsifying only $\psi$ leads to higher or similar performance to sparsifying the whole network.
>
> | Static Sparsity          | Sparsity 80 (%) | Sparsity 95 (%) |
> | :-------------------------- | :-------------: | :-------------: |
> | Both $\psi$ and $\phi$ sparsified |     3.246 (3.08, 3.412)     |  2.444 (2.309,2.578)      |
> | Only $\psi$ sparsified      |      3.245 (3.805, 3.405)      |      3.745   (3.452,4.0372)    |
>
> | Pruning          | Sparsity 80 (%) | Sparsity 95 (%) |
> | :-------------------------- | :-------------: | :-------------: |
> | Both $\psi$ and $\phi$ sparsified |   2.620  (2.499, 2.742)    | 2.558 (2.383, 2.732) |
> | Only $\psi$ sparsified      |    2.399  (2.215, 2.583 )     | 2.573  (2.451, 2.695) |
>
> > **Q3:** The scaling by 4 factor seems arbitrary without further context, though there is likely a good reason for it. Is it related to a specific architectural choice, computational constraint, or a baseline from previous work?
>
> **A:** Our choice of a scaling factor of 4 for most of the analysis is based on the observation found in prior works that 4x provides substantial performance improvements, whereas further scaling offers only marginal benefits [2,4]. Nevertheless we provide our main results for four different scales (Figure 6). We will extend our discussion in Section 3.2 on the justification for this choice.
>
> > **Q4:** Figure 10 & Performance Claims. You conclude that "the performance of the 2 scaled models is nearly identical." However, observation of the IQRs (presumed to be the shaded regions) shows non-overlapping distributions across many frame numbers. If the scaling of this figure were different, these deviations might appear more significant. While the overall conclusion regarding the bottleneck might be correct, it's important to acknowledge and comment on these visible deviations in the IQRs. If the differences are not considered significant, please explain why.
>
> **A:** We agree with the reviewer that the use of “nearly identical” is not precise. We will revise the text to state that the methods achieve 'comparable performance' for greater clarity.
>
> > **Q5:** Overlapping distributions for GAP and SoftMoE are shown [boxplots]. It's questionable whether a notched boxplot or a Wilcoxon rank-sum test would show statistically significant differences in their medians. Given this visual overlap, can you provide a more robust argument for why GAP is considered "higher performing than all other methods." If statistical tests were performed, please give the results. If not, consider performing them or refining the claim.
>
> **A:** We will clarify that with respect to SoftMoE, GAP achieves similar performance but is both more efficient and much simpler.
>
>
> [1] Sokar, Ghada, et al. "The dormant neuron phenomenon in deep reinforcement learning." International Conference on Machine Learning. PMLR, 2023.
>
> [2] Ceron, Johan Samir Obando, Aaron Courville, and Pablo Samuel Castro. "In value-based deep reinforcement learning, a pruned network is a good network." International Conference on Machine Learning. PMLR, 2024.
>
> [3] Graesser, Laura, et al. "The state of sparse training in deep reinforcement learning." International Conference on Machine Learning. PMLR, 2022.
>
> [4] Ceron, Johan Samir Obando, et al. "Mixtures of Experts Unlock Parameter Scaling for Deep RL." International Conference on Machine Learning. PMLR, 2024.

---

> > ### Comment · Reviewer_iWUt · 2025-08-01
> >
> > thanks very much for the full responses, and for doing the extra experiments to validate the approach - it's good to see the method perform well across these. I will be keeping my score at 5.

---

### Official Review · Reviewer_BBqr · 2025-06-29

**Clarity:** 4
**Significance:** 3
**Originality:** 3
**Rating:** 5
**Confidence:** 4

**Summary:**

Scaling up the performance of visual deep reinforcement learning has been a challenging task for long time, where usually scaling up the network lead to worse performance. While many approaches have been proposed in the past, including sparsity and MoE, they are usually complex. This paper identifies the problem that prevents scaling to be the size of the bottleneck layer (the output of encoder). When the size of bottleneck is large (which is usually the case in visual tasks), scaling becomes challenging. The authors introduces global average pooling (GAP) as a simple yet effective approach architectural change to the encoder that reduce the bottleneck size and allow for performance gains.

**Questions:**

In Figure 6, why do we see a performance drop when we scale the width by a multiple of 2 and increase with a multiple of 4?

**Ethical Concerns:**

["NO or VERY MINOR ethics concerns only"]

**Final Justification:**

The authors committed to making the necessary changes in the revised version that will address my concerns.

**Limitations:**

The authors adequately addressed the limitations of their approach in the limitations section.

**Quality:**

3

**Strengths And Weaknesses:**

**Strengths**:
- The paper is easy-to-follow.
- The paper provide a comprehensive set of experiments studying various aspects of the problem and solutions. I enjoyed reading the analysis and discussion.
- The approach is simple yet effective and can be easily incorporated into various approaches which enhance its adoption.
- The method is hyper-parameter free which means it can be included easily into visual RL methods.

**Weaknesses:**
- The width-depth scaling claims is unsupported. The authors claim that GAP unlocks width and depth scaling, but this is never supported with an experiment. I except to see two figures both having performance on the y-axis vs width and depth on the x-axis. The current experiment only shows that the performance using GAP is better than the baseline in wide, deep, and wide&deep variations. No claims about scaling can be made using this experiment. If the authors want to keep the claim, then they need to support it with the proper evidence.
- The authors use the word scaling in a different way from its common use which makes things a bit complicated. The argument of the scaling is almost always missing each sentence. For example, one can argue a method X allows (performance) scaling with the width of the encoder layers then show a figure with the width on the x-axis and performance on the x-axis. This supports the claim clearly. On the other hand, the paper use the word scaling loosely which makes it lose its meaning altogether. The only place where such experiment is presented in Figure 6 (first two) where the performance barely scales with width and doesn't scale. with replay ratio.
- The paper lacks providing any reason for this bottleneck phenomenon. It remains unclear why the bottleneck can severely affect performance. I think the paper should at least give an explanation or even a hypothesis for why this is the case.

I think the paper provides a good contribution, but I think the authors should work on their claims to avoid overstating the results. I think the paper addresses an important problem and there is an audience for it. I highly encourage the authors to improve their writing and moderate their claims to reflect what is given in the experiments.

**Minor issues:**
- "We conduct a series of analyses, both quantitative and qualitative, to provide evidence for our main hypothesis." -> typically the purpose of the experiment is to validate the hypothesis rather than giving evidence for its correctness. I suggest saying "validate our main hypothesis"

---

> ### Author Rebuttal · Authors · 2025-07-31
>
> We thank the reviewer for their feedback! We are glad that the reviewer found our experiments comprehensive and that the method is both effective and readily adaptable.
>
> > **W1:** The width-depth scaling claims is unsupported. The authors claim that GAP unlocks width and depth scaling, but this is never supported with an experiment. I except to see two figures both having performance on the y-axis vs width and depth on the x-axis. The current experiment only shows that the performance using GAP is better than the baseline in wide, deep, and wide&deep variations. No claims about scaling can be made using this experiment. If the authors want to keep the claim, then they need to support it with the proper evidence.
>
> **A:** We have conducted extra experiments with varying depth on Rainbow using the ResNet architecture. Due to the high computational costs and the limited rebuttal period, we report the IQM with 95% confidence score over 20 games at 40M steps. As demonstrated below GAP consistently significantly improves performance of deeper networks. Results across varied width are presented in Figure 6.
>
> | Method   | Deeper-1 | Deeper-2 | Deeper-3 |
> | :------- | :------: | :------: | :------: |
> | Baseline |   2.33 (2.164, 2.496)  |   1.95  (1.89, 2.01)  |   1.63 (1.577,1.683)   |
> | GAP      |   3.47(3.249,3.691)   |   4.10 (3.885, 4.315)  |   3.30 (3.074,3.52 )   |
>
>
> > **W2:** The authors use the word scaling in a different way from its common use which makes things a bit complicated. The argument of the scaling is almost always missing each sentence. For example, one can argue a method X allows (performance) scaling with the width of the encoder layers then show a figure with the width on the x-axis and performance on the x-axis. This supports the claim clearly. On the other hand, the paper use the word scaling loosely which makes it lose its meaning altogether. The only place where such experiment is presented in Figure 6 (first two) where the performance barely scales with width and doesn't scale. with replay ratio.
>
> **A:** In most of our experiments we focus on scaling the width of the dense layers, following previous work [1,2]. In the leftmost plot of Figure 6, GAP consistently outperforms the corresponding scaled baseline. Moreover, the scaled network with GAP outperforms the unscaled baseline. In the second from left plot, a constant scale of 4 is used across varied replay ratios in the x-axis; we show that GAP helps enable obtaining sample efficient agents, as shown by the improved performance over the baseline with the same replay ratio value. We will add a precise definition of scaling and further clarify these results in the revised version of the paper.
>
> > **W3:** The paper lacks providing any reason for this bottleneck phenomenon. It remains unclear why the bottleneck can severely affect performance. I think the paper should at least give an explanation or even a hypothesis for why this is the case.
>
> **A:** We believe that the flattening operation contributes to forming this bottleneck. It transforms the encoder's output into a large 1D vector which, combined with the scaled width of the next layer, creates a high-density connection. This high density makes it difficult to make effective use of the  features' spatial relationships, and may result in overfitting which often causes a loss of plasticity [3]. These effects can result in the model attending to unimportant features and regions in the input, as demonstrated in Figure 3. To understand this further, we show in Figure 4 that preserving the spatial structure by learning higher level features before forwarding them to the dense layers mitigates this issue. We will extend our discussion on this point in the revised version.
>
> > **Q1:** n Figure 6, why do we see a performance drop when we scale the width by a multiple of 2 and increase with a multiple of 4?
>
> **A:** GAP x2 outperforms or matches GAP x1 in 19 out of 20 games. The slight reduction in the aggregated score is due to a single outlier game, breakout, where we observed high variance across our five runs. We will include detailed per-game results in the revised manuscript to make this clear.
>
> > I think the paper provides a good contribution, but I think the authors should work on their claims to avoid overstating the results. I think the paper addresses an important problem and there is an audience for it. I highly encourage the authors to improve their writing and moderate their claims to reflect what is given in the experiments.
>
> **A:** We thank the reviewer for their constructive suggestion and their acknowledgment of the problem's importance and the potential interest our work holds for the community. We agree that our use of the term scaling could be clearer. In the revised manuscript, we will explicitly define it, and incorporate the new experiments presented in this response that further confirm our findings.
>
> > Minor rephrasing
>
> Thank you for providing us with suggestions for rewording, we will incorporate that in the revised manuscript.
>
> [1] Ceron, Johan Samir Obando, et al. "Mixtures of Experts Unlock Parameter Scaling for Deep RL." International Conference on Machine Learning. PMLR, 2024.
>
> [2] Sokar, Ghada, et al. "Don't flatten, tokenize! Unlocking the key to SoftMoE's efficacy in deep RL." The Thirteenth International Conference on Learning Representations.
>
> [3] Nikishin, Evgenii, et al. "The primacy bias in deep reinforcement learning." International conference on machine learning. PMLR, 2022.

---

> ### Comment · Reviewer_BBqr · 2025-08-04
>
> **W1 and W2:** I thank the authors for providing the depth-scaling experiment. This increases my confidence in the claims in the paper. However, one remaining concern is the wording. As you can see in the table provided, the performance of GAP from Deeper-2 to Deeper-3 actually decreases, so it still cannot allow for performance scaling with depth or width (e.g., performance drops when increasing depth or width). In the paper, you should clearly mention this limitation and always say "better scaling than baseline" instead of just "better scaling".
>
>
> I thank the authors for their rebuttal. I think my concerns are addressed, given that the authors promised to make the necessary changes in the revised version. I will change my rating to reflect that.

---

### Official Review · Reviewer_ZiVM · 2025-06-30

**Clarity:** 3
**Significance:** 3
**Originality:** 2
**Rating:** 4
**Confidence:** 5

**Summary:**

Scaling deep reinforcement learning (RL) in vision-based environments has long posed challenges that contrast with the consistent performance gains observed in supervised learning. This paper identifies a key bottleneck in typical pixel-based RL architectures—specifically, the interface between the convolutional encoder (denoted $\phi$) and the subsequent fully connected layers ($\psi$). The authors offer a unifying explanation for the success of recent architectural and algorithmic innovations, arguing that these methods implicitly address this $\phi$ → $\psi$ bottleneck. In response, the paper proposes a principled yet minimal intervention: inserting a Global Average Pooling (GAP) layer to reduce the spatial output of $\phi$ into a compact C-dimensional representation before passing it to $\psi$. This simple modification yields substantial empirical benefits across multiple benchmarks (Atari 200M, Atari 100k, Procgen), enabling both width and depth scaling, improving representational quality (via reduced neuron dormancy and feature norm magnitudes), and offering superior runtime efficiency relative to more complex approaches such as mixtures-of-experts and pruning-based scheme

**Questions:**

1. Have the authors evaluated the effectiveness of GAP in data-augmented actor-critic settings such as DrQ or DrQ-v2? These methods have demonstrated strong performance in pixel-based continuous control tasks and incorporate architectural bottlenecks by design. Notably, the DrQ architecture includes a hard-coded low-dimensional bottleneck, and it is well known within the research community that naively removing this component leads to substantial performance degradation. It would be valuable to position the proposed bottleneck intervention in the context of such existing architectural choices.

2. Figure 4 suggests that scaling both $\phi$ and $\psi$ improves performance, but scaling only $\psi$ hurts performance. Could the authors include results for a $\phi$-only scaling variant to further substantiate the attribution of degradation of only scaling $\psi$? This ablation would help isolate the relative contributions of encoder and head scaling and offer stronger support for the claim that the performance degradation resides primarily in $\psi$

3. The paper focuses heavily on scaling the fully connected head $\psi$, while prior work in computer vision typically emphasizes scaling the encoder $\phi$. Could the authors elaborate on the motivation behind this choice? While it is understood that, in typical vision-based RL architectures, $\psi$ contains the majority of the parameters, it remains unclear why encoder scaling—which is the standard approach in computer vision tasks—is deprioritized. From a representational learning perspective, enhancing the encoder capacity would seem more intuitively aligned with performance improvements.

4. Given that GAP aggregates over spatial dimensions, how do the authors view its applicability in environments that require fine-grained spatial reasoning or partial observability? For instance, continuous control domains such as Adroit or the DeepMind Control Suite may suffer from the loss of spatial information. Have the authors explored or considered this trade-off?

5. Minor typo – “Procegn → Procgen” in Fig 11

**Ethical Concerns:**

["NO or VERY MINOR ethics concerns only"]

**Final Justification:**

Many of my concerns have been addressed through the additional experimental results provided by the authors, which I appreciate. However, I remain somewhat unconvinced about the underlying motivation and the reasons behind the effectiveness of the proposed approach. Therefore, I am adjusting my score to Borderline Accept, rather than Accept.

**Limitations:**

yes

**Quality:**

3

**Strengths And Weaknesses:**

Strengths

- The paper proposes a simple yet highly effective architectural modification—Global Average Pooling (GAP)—to address a longstanding challenge in scaling vision-based deep RL.
- The authors present a unifying perspective suggesting that the performance improvements observed in recent methods such as SoftMoE and pruning primarily stem from their implicit targeting of the architectural bottleneck—namely, the connection between the encoder $\phi$ and the head $\psi$. This insight provides a valuable conceptual lens for understanding scaling dynamics in vision-based deep RL. Moreover, the paper offers empirical evidence that effective depth scaling is possible—an underexplored axis of architectural scaling, as prior work has predominantly focused on width alone.
- The study features extensive empirical validation, including large-scale experiments on the full Arcade Learning Environment (ALE) with 200 million frames, the 100k ALE setting, and Procgen, thereby demonstrating the robustness and generality of the proposed method.
- The proposed approach is not only performant but also computationally efficient and implementation-friendly, requiring minimal architectural changes and providing faster training time relative to alternatives such as SoftMoEs and pruning based methods.


Weaknesses

- While the paper presents compelling empirical evidence for the role of the $\phi$ → $\psi$ bottleneck in scaling performance, its explanation remains largely empirical. The authors do not provide a theoretical account or a detailed representational analysis to clarify why the absence of a well-structured bottleneck—and the use of flattened encoder outputs—leads to degraded plasticity or impaired learning dynamics. Given that this phenomenon lies at the core of the paper’s contribution, a more principled explanation would substantially strengthen the work’s impact and conceptual clarity.
- Despite broad benchmarking, the experimental domains remain confined to pixel-based benchmarks with discrete action spaces (e.g., ALE and Procgen), with no evaluation on continuous control domains (e.g., DeepMind Control Suite, Adroit), limiting the scope of generalizability. In addition, I think that
- The absence of parameter count reporting presents a potential concern. Since the introduction of the GAP layer substantially reduces the number of parameters by collapsing spatial dimensions, it remains unclear whether the observed performance gains result from genuine improvements in scaling capacity or merely from more efficient parameter utilization. In typical scaling discussions, increased model size is often implied; however, with GAP, the networks may actually become smaller. Clarifying this distinction—and including performance comparisons at matched parameter counts—would offer a more rigorous understanding of whether GAP enables true architectural scaling or simply more effective use of fewer parameters.

---

> ### Author Rebuttal · Authors · 2025-07-31
>
> We thank the reviewer for their feedback! We are glad that the reviewer found our empirical validation to be extensive and the approach highly effective, computationally efficient, implementation-friendly, and requiring minimal architectural changes. Below, we provide detailed responses to each comment, incorporating the results from all requested experiments.
>
> > **W1:** why the absence of a well-structured bottleneck—and the use of flattened encoder outputs—leads to degraded plasticity or impaired learning dynamics…. a more principled explanation would substantially strengthen the work’s impact and conceptual clarity.
>
> **A:** The flattening operation transforms the encoder's output into a large 1D vector, potentially losing some of the spatial structure captured by the encoder. This flattened vector, combined with the scaled width of the next layer, creates a high-density connection, which makes it difficult to make effective use of the  features' spatial relationships, and may result in overfitting which often causes a loss of plasticity [1]. These effects can result in the model attending to unimportant features and regions in the input, as demonstrated in Figure 3. To understand this further, we demonstrated in Figure 4 that preserving the spatial structure by learning higher level features before forwarding them to the dense layers mitigates this issue.
>
> > **W2, Q4:** “Despite broad benchmarking, the experimental domains remain confined to pixel-based benchmarks with discrete action spaces (e.g., ALE and Procgen), with no evaluation on continuous control domains (e.g., DeepMind Control Suite, Adroit), limiting the scope of generalizability.” “how do the authors view its applicability in environments that require fine-grained spatial reasoning or partial observability?”
>
> **A:** Following the reviewer’s suggestion, we expand our evaluation to the continuous control domain. We evaluate SAC on DeepMind Control tasks. We ran each experiment with 5 seeds. We follow the CNN architecture presented in [1], which consists of four convolutional layers, followed by critic and actor embeddings of a size of 50 neurons.  We scale the critic and actor embedding by 8x and 16x. GAP consistently yields significant performance gains over its corresponding baseline.
>
> | Method       | cartpole-swingup    | cheetah-run         | walker-walk         | walker-stand        |
> |:-------------|:-------------------:|:-------------------:|:-------------------:|:-------------------:|
> | Baseline x8  | 512.41 ± 127.26     | 116.71 ± 94.86      | 55.85 ± 41.89       | 190.43 ± 37.63      |
> | GAP x8       | **839.03 ± 15.42**      | **389.68 ± 61.69**      | **690.53 ± 83.53**      | **938.62 ± 29.06**      |
>
> | Method       | cartpole-swingup    | cheetah-run         | walker-walk         | walker-stand        |
> |:-------------|:-------------------:|:-------------------:|:-------------------:|:-------------------:|
> | Baseline x16 | 558.57 ± 238.98     | 171.13 ± 93.32      | 119.39 ± 77.88      | 174.69 ± 27.76      |
> | GAP x16      | **830.64 ± 15.70**      | **349.97 ± 74.77**      | **545.89 ± 42.81**      | **952.04 ± 15.28**      |
>
> We thank the reviewer for pressing us on this point, as the addition of these extra results help reinforce the generality of our findings.
>
>
> > **W3:** The absence of parameter count ….it remains unclear whether the observed performance gains result from genuine improvements in scaling capacity or merely from more efficient parameter utilization. Clarifying this distinction—and including performance comparisons at matched parameter counts—would offer a more rigorous understanding of whether GAP enables true architectural scaling or simply more effective use of fewer parameters.
>
> **A:** We reported the parameter count in Figure 5, right. Note that the parameter count of GAP matches the parameter count of SoftMoE-1 (Please see Figure 1 for illustration and Section 2 for more details). We also tested a baseline with this same parameter count. As demonstrated in Figure 5, parameter-matched baseline performs worse than other methods, confirming the performance improvements are attributable to our architectural design, not just parameter count.
>
> > **Q1:** Have the authors evaluated the effectiveness of GAP in data-augmented actor-critic settings such as DrQ or DrQ-v2?
>
> **A:** Following the reviewer's suggestion, we ran additional experiments using the DrQ (top) and DrQ-v2 (bottom) algorithms on the Atari100K benchmark with a 4x scaled network using 5 seeds. Consistent with previous findings, the results show that GAP provides performance gains in both cases.
>
> | DrQ   | CrazyClimber            | Alien              | Gopher             | MsPacman           |
> |:---------|:-----------------------:|:------------------:|:------------------:|:------------------:|
> | Baseline | 9,429.90 ± 1,138.75   | 488.95 ± 24.72     | 296.92 ± 30.44     | 663.18 ± 39.88     |
> | GAP      | **11,950.33 ± 533.28**    | **568.34 ± 8.00**      | **350.05 ± 17.22**     | **764.94 ± 18.83**     |
>
> | DrQ-v2   | CrazyClimber            | Alien              | Gopher             | MsPacman           |
> |:---------|:-----------------------:|:------------------:|:------------------:|:------------------:|
> | Baseline | 7,971.73 ± 2,800.05   | 498.15 ± 22.12     | 198.44 ± 24.24     | 725.83 ± 35.85     |
> | GAP      | **12,947.54 ± 812.50**    | **570.60 ± 18.08**     | **320.22 ± 10.47**     | 753.51 ± 41.21     |
>
> We once again thank the reviewer for suggesting this extra set of experiments.
>
>
> > **Q2:** Figure 4 suggests that scaling both $\phi$ and $\psi$ improves performance, but scaling only $\psi$ hurts performance. Could the authors include results for a $\phi$-only scaling variant to further substantiate the attribution of degradation of only scaling $\psi$? This ablation would help isolate the relative contributions of encoder and head scaling and offer stronger support for the claim that the performance degradation resides primarily in $\psi$
>
> **A:** As suggested by the reviewer, we conducted an experiment with a deeper $\phi$ while maintaining the original, unscaled size for the $\psi$. We report the average return at 100M steps. As shown below, providing high-level features to $\psi$ enhances performance in both its unscaled and scaled versions, with the most significant gains observed in the scaled version.
>
> | Model                  | Asterix                    | Seaquest                   | DemonAttack                | SpaceInvaders              |
> |:----------------------------|:--------------------------:|:--------------------------:|:--------------------------:|:--------------------------:|
> | Baseline                    | 40,107.83 $\pm$ 9,727.30     | 117,374.13 $\pm$ 92,323.04   | 67,238.46 $\pm$ 11,705.80    | 13,490.66 $\pm$ 6,035.76     |
> | Deeper $\phi$                 | 75,044.80 $\pm$ 16,696.87    | 249,930.37 $\pm$ 48,719.17   | 60,223.30 $\pm$ 20,860.26    | 12,531.67 $\pm$ 2,569.25   |
> | Scaled $\psi$                 | 14,562.03 $\pm$ 2,652.72     | 33,940.98 $\pm$ 25,256.58    | 8,053.09 $\pm$ 673.43        | 9,946.47 $\pm$ 2,244.52      |
> | Deeper $\phi$ + Scaled $\psi$ | 127,310.99 $\pm$ 12,471.02   | 230,295.82 $\pm$ 92,301.67   | 70,754.62 $\pm$ 21,999.48    | 15,556.92 $\pm$ 2,924.93     |
>
>
> > **Q3:** The paper focuses heavily on scaling the fully connected head , while prior work in computer vision typically emphasizes scaling the encoder . Could the authors elaborate on the motivation behind this choice? While it is understood that, in typical vision-based RL architectures,  contains the majority of the parameters, it remains unclear why encoder scaling—which is the standard approach in computer vision tasks—is deprioritized. From a representational learning perspective, enhancing the encoder capacity would seem more intuitively aligned with performance improvements.
>
> **A:** Our findings in Figure 2, right, demonstrate that scaling the dense layers alone leads to a performance decline comparable to that of scaling both the encoder and dense layers simultaneously. This observation, along with the success of recent methods like SoftMoE—which achieved optimal performance by applying mixtures to the dense layers—motivates our focus in this paper on scaling $\psi$. Nevertheless, we conducted an additional experiment scaling only the width of the encoder layers by 4x. Due to the high computational costs and the limited rebuttal period, we report the IQM with 95% confidence interval over 20 games at 40M steps. Consistent with our previous findings, scaling the encoder decreases the performance of the model, whereas incorporating GAP into the network leads to a significant improvement.
>
> | Model             | IQM (CIs) |
> |:-----------------------|:-------|
> | Baseline               | 3.0327 (2.847, 3.2184) |
> | Scaled $\phi$          | 2.6721 (2.55, 2.7942) |
> | Scaled $\phi$ w/ GAP | 5.0173 (4.817, 5.2176) |
>
>
> [1] Nikishin, Evgenii, et al. "The primacy bias in deep reinforcement learning." International conference on machine learning. PMLR, 2022.
>
> [2] Yarats, Denis, et al. "Improving sample efficiency in model-free reinforcement learning from images." Proceedings of the aaai conference on artificial intelligence. Vol. 35. No. 12. 2021.

---

> > ### Comment · Reviewer_ZiVM · 2025-08-07
> >
> > 1. I’m a bit confused by the statement that Global Average Pooling (GAP) preserves spatial structure. My understanding is that GAP, by design, collapses spatial dimensions and is generally considered one of the most aggressive ways of discarding spatial information. Could you clarify this point?
> >
> > 2. It’s encouraging to see that GAP also performs well on continuous control tasks such as DMC. I appreciate the effort to run these additional experiments.
> >
> > 3. Thanks for the clarification on the earlier point.
> >
> > 4. Once again, thank you for providing the additional experimental results—they help strengthen the paper.
> >
> > 5. The result that scaling \phi seems to be beneficial is quite interesting. This finding is intriguing and potentially valuable for future work. Thank you for including this analysis.
> >
> > 6. I apologize if my earlier question was unclear. My intent was to understand the rationale behind scaling the fully connected (FC) layer rather than the encoder. In recent literature on scaling in vision-based reinforcement learning, there has typically been a focus on scaling the final layers, though there is growing interest in scaling the encoder itself. From my perspective, it would be helpful to understand whether the authors believe that scaling the final layer is inherently more impactful than scaling the encoder. This seems somewhat at odds with common trends in computer vision, where most scaling efforts focus on earlier parts of the network rather than the final layer.
> >
> >
> > Overall, I feel that the authors have addressed most of the concerns raised in my initial review. I appreciate their efforts and, based on the revisions and clarifications provided, I am raising my score to Borderline Accept.

---

> > > ### Author Response · Authors · 2025-08-07
> > >
> > > We thank the reviewer for their time reading our response and reconsidering their score. We are glad that they found the results of the suggested experiments strengthened the paper. Below we provide clarification to the last two points.
> > >
> > > 1.We agree with the reviewer that compared to more granular structure like per-patch or per-conv tokens [1], GAP is a more aggressive compression technique that could lose some spatial structure. Nevertheless, it still maintains the spatial information in each feature map more effectively than the flattening operation. We appreciate you pointing this out and we will provide further clarification in the revised version.
> > >
> > > 6.Thanks for the insightful question. We believe that studying both directions are valuable and important. As part of the paper’s contribution is understanding the reasons behind the success of previous methods, we focused on scaling dense layers. We agree with the reviewer that understanding the challenges in scaling the encoder is a valuable research direction. We believe the promising results of the experiment you suggested would encourage future work in this area.
> > >
> > > [1] Sokar, Ghada, et al. "Don't flatten, tokenize! Unlocking the key to SoftMoE's efficacy in deep RL." The Thirteenth International Conference on Learning Representations.

---

### Official Review · Reviewer_YprL · 2025-07-06

**Clarity:** 3
**Significance:** 3
**Originality:** 2
**Rating:** 3
**Confidence:** 4

**Summary:**

This paper considers the problem of scaling in pixel-based deep RL, that is, under what circumstances is it possible to improve the performance of deep RL algorithms by increasing the size of the network. They find that using global average pooling after the convolutional layers allows scaling. They show good empirical performance with this simple idea on Atari benchmarks.

**Questions:**

* Fig. 2 (Right) which game is this?

**Ethical Concerns:**

["NO or VERY MINOR ethics concerns only"]

**Final Justification:**

I find it rather strange that the very standard (in CV) global average pooling operation has not be tried in deep RL and that this is now a discovery. But maybe it really is because I haven't seen any papers on this. They provide no sound rationale for why GAP allows scaling. I find the general explanations in the rebuttal and paper to be imprecise and not compelling. The experiments also could have compared to a wider range of previous works (e.g., MOE) instead of only comparing to a single "baseline" selected by the authors. The plots and figures are somewhat empty and give me the impression of an unfinished research project.

There is not really any discussion of RL in the paper. What's observed in the paper is related to representation learning and I suspect one should be able to see something similar in image classification (but perhaps less pronounced). This would be an important experiment to add and if the paper is accepted I kindly ask the AC to suggest this experiment to the authors (my bad for not doing so in my review).

I maintain my score but I'm OK with democratically accepting the paper since other reviewers have a more positive opinion of the paper.

**Limitations:**

Yes.

**Paper Formatting Concerns:**

None.

**Quality:**

2

**Strengths And Weaknesses:**

## Strengths

* The paper achieves strong performance with increasing model size and a simple architectural modification: global average pooling after convolutional layers in the value network.

## Weaknesses

* The main hypothesis of the paper says low-dimensional and well structured bottleneck allows scaling. Does that mean that a 2x2xd bottleneck doesn't allow scaling? Also, what does well-structured really mean? Does it have a clear definition? From my understanding of the paper it's only shown that global average pooling allows scaling but then why turn this into an unclear and more general statement?
* The paper is technically void. There's not a single new or interesting equation (let alone theorem) in the paper.
* The point of the paper could have been conveyed in one page.

**Minor:**
An expectation is missing in the definition of Q-value.

**Summary:**
On the one hand, the finding is interesting and can have impact, on the other hand, the paper is bloated and has the weaknesses I've mentioned. Therefore, I am borderline on this paper.

---

> ### Author Rebuttal · Authors · 2025-07-31
>
> We thank the reviewer for their feedback, and we appreciate their recognition of the method's strong performance, simplicity, and impact. Below we provide responses to all comments.
>
> > **W1:** The main hypothesis of the paper says low-dimensional and well structured bottleneck allows scaling. Does that mean that a 2x2xd bottleneck doesn't allow scaling? Also, what does well-structured really mean?
>
> **A:** What we mean is organizing the encoder's output in a way that facilitates feature learning in the dense layers; maintaining the features' spatial relationships and learning more generalizable representation. Methods such as sparsity, tokenization, or GAP create a more effective structure than simply flattening the output and densely connecting all neurons. Note that it is not just a matter of parameter count. As we demonstrate in Figure 5 (right), an equally-parameterized baseline still underperforms.
>
> > **W2:** The paper is technically void. There's not a single new or interesting equation (let alone theorem) in the paper.
>
> **A:** While our work doesn't introduce new equations given its empirical nature, we respectfully disagree that this implies a lack of technical depth. The paper provides rigorous analysis, a novel and effective architectural design, and extensive validation of its efficiency across diverse settings and metrics. Further, we demonstrate that one can reduce the complexity of prior approaches (such as MoEs and tokenization) without sacrificing accuracy. We strongly believe this type of evidence-backed insight is a valid contribution to the field, and particularly in light of its simplicity.
>
> > **W3:** The point of the paper could have been conveyed in one page.
>
> **A:** The paper's value extends beyond our final finding of the architectural modification that boosts the agent’s performance. Indeed, a core part of our work is the extensive set of analyses which provide the evidence for our final claim. Without these, the paper would reduce to an anecdotal observation, rather than useful insights backed by data and analyses.
>
> > **Q1:** Fig. 2 (Right) which game is this?
>
> **A:** It is aggregated results over the same set of 20 games used in all our experiments. We will clarify this in the caption.

---

> > ### Comment · Reviewer_YprL · 2025-08-04
> >
> > I thank the authors for their response. I don't believe that my question regarding their "main hypothesis" has been addressed. "well-structured" is not a technical term. Also, parts of the question where entirely ignored. I recommend the authors take more care in writing their main hypothesis or leave it out entirely in future revisions.

---

> > > ### Author Response · Authors · 2025-08-04
> > >
> > > We thank the reviewer for their response and pushing for greater clarity! We will address each part in turn to provide a comprehensive clarification.
> > >
> > > > Does that mean that a 2x2xd bottleneck doesn't allow scaling?
> > >
> > > The ability to scale can be made possible by learning high-level features that effectively compress and abstract information from lower-level features (e.g. pixel inputs) into a compact representation, such as 2x2xd, that facilitates feature learning in the dense layers. We demonstrated this principle in our Figure 4 analysis, where the encoder produces a compact 3x3xd feature map (with d=32).
> > >
> > > > What does well-structured really mean?
> > >
> > > The flattening operation in the baseline network transforms the encoder's output into a large 1D vector, potentially losing some of the spatial structure captured by the encoder. This flattened vector, combined with the scaled width of the next layer, creates a high-density connection, which may make it difficult to make effective use of the features' spatial relationships. We consider this flattened representation ‘unstructured’. Prior work [1,2] has shown that structuring the encoder’s output by organizing it in a way that facilitates feature learning rather than flattening (in the form of experts in [1] and tokens in [2]) enables scaling in deep RL networks. Our paper extends this line of work with global average pooling, which we consider an alternate form of structure.
> > >
> > >
> > > We agree with the reviewer that the terminology is not a common technical term in deep RL literature. We will be happy to add the discussion above to the final version to help clarify to readers what is meant by "structured". If the reviewer has further suggestions for improving this point, we would be happy to incorporate them.
> > >
> > > We thank the reviewer for pressing us on this point, as it helps improve the clarity of our work.
> > >
> > >
> > > [1] Ceron, Johan Samir Obando, et al. "Mixtures of Experts Unlock Parameter Scaling for Deep RL." International Conference on Machine Learning. PMLR, 2024.
> > >
> > > [2] Sokar, Ghada, et al. "Don't flatten, tokenize! Unlocking the key to SoftMoE's efficacy in deep RL." The Thirteenth International Conference on Learning Representations.

---

### Decision · Program_Chairs · 2025-09-17

**Decision:**

Accept (poster)

**Comment:**

This paper investigates the challenges that lead to performance degradation in scaled RL networks and analyzes the reasons behind the success of existing architectural approaches to scaling. Building on these insights, it presents pooling as a faster, simpler method that yields superior performance.

All reviewers acknowledged the importance of the problem and appreciated the simplicity and effectiveness of the proposed pooling (GAP) strategy in mitigating scalability bottlenecks. While concerns were raised regarding the lack of a strong theoretical explanation, the AC notes that in reinforcement learning it is often difficult or even artificial to provide such justifications.

The AC concurs with the reviewers and recommends acceptance of this paper.